# 1 Modeling water balance using the Budyko framework over variable

## 2 timescales under diverse climates

- 3 Chuanhao Wu<sup>1</sup>, Pat J.-F. Yeh<sup>2</sup>, Kai Xu<sup>1</sup>, Bill X. Hu<sup>1\*</sup>, Guoru Huang<sup>3,4</sup>, and Peng Wang<sup>1,5</sup>
- <sup>4</sup> <sup>1</sup>Institute of Groundwater and Earth Sciences, Jinan University, Guangzhou 510632, China.
- <sup>2</sup> Department of Civil and Environmental Engineering, National University of Singapore, Singapore.
- <sup>3</sup>School of Civil Engineering and Transportation, South China University of Technology, Guangzhou 510640, China.
- <sup>4</sup> State Key Laboratory of Subtropical Building Science, South China University of Technology, Guangzhou 510640, China.
- <sup>5</sup> Chongqing Key Laboratory of Karst Environment, Chongqing 400715, China.

9 Correspondence to: Bill X. Hu (bill.x.hu@gmail.com)

## 10 Abstract

11 Understanding the effects of climate and catchment characteristics on overall water balance at different temporal scales 12 remains a challenging task due to the large spatial heterogeneity and temporal variability. Based on a long-term (1960–2008) 13 land surface hydrologic dataset over China, this study presented a systematic examination of the applicability of the Budyko 14 model (BM) under various climatic conditions at long-term mean annual, annual, seasonal and monthly temporal scales. The 15 roles of water storage change (WSC, dS/dt) in water balance modeling and the dominant climate control factors on modeling 16 errors of BM are investigated. The results indicate that BM performs well at mean annual scale and the performance in arid 17 climates is better than humid climates. At other smaller timescales, BM is generally accurate in arid climates, but fails to 18 capture dominant controls on water balance in humid climates due to the effects of WSC not included in BM. The accuracy 19 of BM can be ranked from high to low as: dry seasonal, annual, monthly, and wet seasonal timescales. When WSC is 20 incorporated into BM by replacing precipitation (P) with effective precipitation (i.e., P minus WSC), significant 21 improvements are found in arid climates, but to a lesser extent in humid climates. The ratio of the standard deviation of WSC 22 to that of evapotranspiration (E), which increases from arid to humid climates, is found to be the key indicator of the BM 23 simulation errors due to the omission of the effect of WSC. The modeling errors of BM are positively correlated with the 24 temporal variability of WSC and hence larger in humid climates, and also found to be proportional to the ratio of potential 25 evapotranspiration (PET) to E. More sophisticated models than the BM which explicitly incorporate the effect of WSC are 26 required to improve water balance modeling in humid climates particularly at all the annual, seasonal, and monthly 27 timescales.

Key words: water balance; Budyko framework; evapotranspiration; water storage change; China

#### 29 1 Introduction

Precipitation (P) partitioning into evapotranspiration (E), runoff (R) and soil water storage is controlled by climate and 31 catchment characteristics, such as soil, topography, and vegetation (Zhang et al., 2008). Quantifying the effects of climate 32 and catchment characteristics on water balance is a major scientific challenge for hydrologists. Budyko (1958) postulated 33 that the first-order control on the partitioning of P is the balance between available water (P) and energy (represented by 34 potential evapotranspiration, PET). The empirical relationship proposed, widely known as the Budyko's curve, has shown 35 good agreement with long-term water balance data for numerous watersheds globally (Budyko, 1974). Based on the Budyko 36 hypothesis, various functional forms associated with the relation between the evaporation ratio (E/P) and climate aridity 37 index (PET/P) have been developed for quantifying long-term water balance (Pike, 1964; Fu, 1981; Choudhury, 1999; Zhang 38 et al., 2001, 2004; Porporato et al., 2004; Yang et al., 2008; Gerrits et al., 2009). The parameters of the analytically derived 39 Budyko-type equations (e.g., Fu, 1981; Zhang et al., 2004; Yang et al., 2008) well account for the effects of rainfall 40 seasonality and soil water storage capacity (Milly, 1994a, 1994b; Potter et al., 2005; Hickel and Zhang, 2006; Yokoo et al., 41 2008; Gerrits et al., 2009; Feng et al., 2012) and vegetation dynamics (Zhang et al., 2001; Donohue et al., 2007; Oudin et al., 42 2008; Yang et al., 2009; Peel et al., 2010; Li et al., 2013; Zhang et al., 2016) on the long-term mean water-energy balance. 43 The Budyko framework has been considered as a useful tool to investigate the interaction between climate, the hydrologic 44 cycle, and catchment characteristics under the steady state at the catchment scale (Donohue et al., 2007, 2010, 2011; Yang et 45 al., 2009; Roderick and Farquhar, 2011; Wang and Hejazi, 2011; Yang and Yang, 2011; Li et al., 2013; Zhang et al., 2016).

Recently numerous research efforts have been directed to extend the applicability of the original Budyko framework to 48 smaller temporal scales than the long-term average, such as annual (Koster and Suarez, 1999; Sankarasubramanian and 49 Vogel, 2002; Zhang et al., 2008; Yang et al., 2007, 2009; Potter and Zhang, 2009; Cheng et al., 2011; Tekleab et al., 2011; 50 Carmona et al., 2014; Yu et al., 2013), seasonal (Chen et al., 2013), monthly (Zhang et al., 2008; Tekleab et al., 2011; Du et 51 al., 2016), and daily (Zhang et al., 2008) timescales. For example, Koster and Suarez (1999) proposed an analytical

framework based on the Budyko equation to quantify the interannual variability of E, and compared the theoretical 53 framework with the global climate model simulations. Zhang et al. (2008) extended the Budyko framework to predict water 54 balance at the mean annual, monthly, and daily timescales in Australia. The results indicated that the models perform 55 well in most catchments in Australia at the mean annual and annual timescales, but more complicated models are required at 56 the shorter timescales primarily due to the importance of water storage change (WSC, dS/dt) at these smaller timescales. 57 Yang et al. (2009) incorporated the impact of vegetation coverage into the Budyko framework to improve parameter 58 estimation in the coupled water-energy balance equation. Chen et al. (2013) extended the Budyko framework to the seasonal 59 timescale and also developed a model for incorporating interannual variability of E and WSC. The WSC has been found to be 60 a significant component on water balance variability at annual or smaller timescales in many regions around the world 61 (Eltahir and Yeh, 1999; Yeh and Famiglietti, 2008; Yokoo et al., 2008; Istanbulluoglu et al., 2012; Wang and Alimohammadi, 62 2012; Wang, 2012; Chen et al., 2013; Du et al., 2016). Wu et al. (2017) investigated the effects of climate and WSC on the 63 temporal variability of E over China. They found that the effect of WSC in accommodating climatic fluctuations is larger at 64 monthly timescale than at annual timescale, and even becomes the dominant controlling factor on the temporal variability of 65 E in some extremely arid regions. Therefore, when applying to smaller timescales, the Budyko framework should be 66 extended to incorporate the effect of WSC (Zhang et al., 2008; Zeng and Cai, 2015). Wang (2012) suggested that the 67 effective precipitation (i.e., P minus WSC) can be used in Budyko's framework to account for the storage change and satisfy 68 the water balance under the non-steady-state condition.

Although advances have been made in water balance simulations over variable timescales within the Budyko framework, our knowledge about the effects of climate and catchment characteristics on the water balance is still limited due to its spatial heterogeneity and temporal variability. The research question asked in this study is whether the Budyko framework is applicable for water balance modeling at smaller timescales when *WSC* becomes increasingly significant, and under various hydroclimate conditions particularly for the extremely humid or arid climates. On the basis of a long-term (1960–2008) gridded land surface dataset over China, the objectives here are: (1) to test the Budyko framework under various climatic conditions over four different timescales (mean annual, annual, seasonal, and monthly) at both the basin and the grid scales;

(2) to investigate the role of *WSC* in water balance modeling at various timescales and under different climate conditions; and (3) to identify the dominant climatic controlling factors on the modeling errors of the Budyko framework. It is anticipated that the knowledge obtained from this study would enhance understanding of the controls of climatic and catchment characteristics on the water balances from arid to humid climates and across a variety of timescales.

#### 81 2 Methodology

#### 82 **2.1 Data sources**

A long-term (1960–2008) meteorological and hydrologic gridded dataset, covering the entire China with a 0.25 ° spatial 84 resolution, is provided by Zhang et al. (2014). The observed meteorological data, including P, maximum temperature  $T_{\text{max}}$ , minimum temperature  $T_{min}$  and wind speed WS, are derived from the total 756 monitoring stations maintained by the Chinese 85 Meteorological Administration (CMA). The hydrologic flux data including E, surface runoff  $(R_s)$  and baseflow  $(B_s)$ , are 86 87 obtained from the Variable Infiltration Capacity (VIC) model simulation forced by daily observed climate forcing data of P, 88  $T_{\text{max}}$ ,  $T_{\text{min}}$ , and WS. The dataset used here provides a more reliable estimate of R, E and other hydrologic variables over China 89 compared with the global products of a similar nature (e.g., Nijssen et al., 2001; Adam et al., 2006; Rodell et al., 2004; 90 Sheffield et al., 2006; Sheffield and Wood, 2007; Pan et al., 2012), and hence it will be treated and termed as the "observed 91 data" in this paper, particularly in contrast to the "predicted data" from the Budyko models. This dataset has been 92 successfully used for the quantitative assessment of effects of climate and WSC on the temporal variability of actual 93 evapotranspiration in China (Wu et al., 2017). The PET data spanning the period 1960–2008 with a 1 °resolution is provided 94 by the Hydroclimatology Group of Princeton University (Sheffield et al., 2006, 2012). The PET is estimated based on the 95 Penman equation (Penman, 1948; Shuttleworth, 1993) using the updated meteorological dataset obtained from Sheffield et al. 96 (2006, 2012).

All the data used are disaggregated into  $0.5^{\circ} \times 0.5^{\circ}$  grids over China (for a total of 3814 grids) using the linear interpolation. Gridded *WSC* at the time interval  $dt_i$  is computed using the water balance equation:

$$dS_i / dt_i = P_i - E_i - R_{si} - B_{si}$$
(1)

A total of 14 large river basins in China under a large diversity of climates, are chosen for analysis (Figure 1(a)). The monthly and annual P, E,  $R_s$ ,  $B_s$  and PET for each of the 14 basins are estimated from the grid data within the basin boundary. The basic and detailed information of the 14 basins and their long-term (1960–2008) mean water balances are given in Table 1. As seen, the mean annual aridity index (PET/P) is less than 1.0 in the Southeast Drainage, Pearl and Yangtze River basins belonging to humid climates, and in the range of 1.19–8.63 for the other eleven basins belonging to arid climates. The net primary productivity (NPP) is an important component of the global carbon budget directly reflecting the production

capacity of vegetation, and it is the basis of organic material and energy cycles in the global ecosystem (Field et al. 1998; 109 Houghton et al. 1999; Field, 2001). The formation of NPP can effectively indicate the ecological response of catchment 110 characteristics. In this study, the 1-km NPP data set in 2010 covering the entire China as obtained from the Research Center 111 for Natural resources and Environment Science Data of the Chinese Academy of Science, is used to represent the vegetation 112 characteristics of catchments over China. Figure 1(b) show the spatial distribution of NPP over China. The value of NPP 113 generally decreases from southern to northwest China, which is generally opposite to the pattern of mean annual aridity 114 index. Scatterplot of NPP versus the mean annual aridity index for the 3814 grids of China is shown in Figure 1(c). As seen, 115 the NPP decreases rapidly with the increasing aridity index for all regions with the aridity index smaller than 3. However, 116 under the extremely dry conditions (the aridity index > 3) the NPP falls within a narrow range fluctuating around zero. NPP 117 is negligible in the extremely dry regions such as some regions of Xinjiang.

## 118 **2.2 Dry and wet seasons**

For a given catchment, the dry and wet months among the twelve months of a year are identified based on the mean monthly aridity index expressed as (Chen et al., 2013)

$$A_{m} = \frac{\overline{PET_{m}}}{\overline{P_{m}} - \frac{\overline{dS_{m}}}{dt_{m}}}$$
(2)

where  $\overline{PET_m}$ ,  $\overline{P_m}$  and  $\frac{\overline{dS_m}}{dt_m}$  are the 1960–2008 monthly climatology of potential evaporation, precipitation and storage

change, respectively. The classification between dry and wet months are based upon the criterion of  $A_m \ge 1$  (dry) or  $A_m 

143 precipitation  $P_{e}$ , (i.e.,  $P_{e} = P - dS / dt$ ), thus Equation (4) becomes (Zeng and Cai, 2015)

144 
$$\frac{E}{P_e} = 1 + \frac{PET}{P_e} - [1 + (\frac{PET}{P_e})^{\omega}]^{1/\omega}$$
(5)

145 In this study, both equations (4) and (5) will be applied to the smaller timescales (i.e. annual, seasonal and monthly). The

146 parameters  $\omega$  is estimated by using the least-square method.

## 147 **2.4 Model performance evaluation**

148 The model performance is evaluated by using the Nash-Sutcliffe Coefficient of Efficiency (*CE*) and the Relative Error (*RE*).

149 The *CE* is calculated as

150 
$$CE = 1 - \frac{\sum_{i=1}^{N} (E_{o,i} - E_{s,i})^2}{\sum_{i=1}^{N} (E_{o,i} - \overline{E}_{o,i})^2}$$
(6)

where  $E_{o,i}$  and  $E_{s,i}$  are the observed and simulated *E* at the *i*th time step, respectively. *N* is the number of the time steps. *CE* ranges from  $-\infty$  to 1. A positive *CE* value indicates acceptable model performance, and a *CE* closer to 1 indicates more efficient model performance (Moriasi et al., 2007).

154

155 The RE is a percentage measurement of mean bias of model simulations relative to observations, defined as

156 
$$RE = \frac{\overline{E}_{s,i} - \overline{E}_{o,i}}{\overline{E}_{o,i}} \times 100\%$$
(7)

157 where  $\overline{E}_{o,i}$  and  $\overline{E}_{s,i}$  represent the mean observed and simulated *E* over the *N* time steps.

#### 158 3 Results

## 159 **3.1 Mean annual water balance**

- 160 **3.1.1 Basin scale**
- 161 Figure 2 (a) shows the mean annual *E/P* versus *PET/P* over the 14 river basins in China. As seen, the BM (equation (4)) with

a fitted parameter  $\omega$  of 1.92 adequately captures the variations in long-term water and energy balances of these basins (Figure 2a). Figure 2(c) shows the comparison between the "observed" and the BM-predicted mean annual *E* in the 14 river basins. Results indicate that the estimated *E* using the Fu's equation agrees well with observations at the long-term mean annual timescale with the correlation coefficient (*r*) of 0.97 and the *RE* of -2.9%. In addition, the estimated mean annual *E* under dry climates is more accurate than that under humid climates.

## 167 **3.1.2 Grid scale**

Figure 2(b) plots the mean annual *E/P* versus *PET/P* along with the distribution of NPP, while Figure 2(d) shows the comparison between observed and BM-simulated mean annual *E*. Both subplots include all the total 3814 grids in China. It is clear that most of grids follow the Budyko curve reasonably well with a fitted  $\omega$  of 1.97, and the aridity index increases with the decreasing NPP. The high *r* (~0.94, Figure 2(d)) indicates that the BM can be calibrated and simulates mean annual water balance well for most grids. Consistent with the above, it is found that the model performs more accurately in the grids with a larger aridity index (i.e. smaller NPP) than more humid grids.

#### 174 **3.2 Annual water balance**

#### 175 **3.2.1 Basin scale**

The BM is also applied to annual water balance modeling at the basin scale. The comparison of observed and BM-simulated 177 annual E in the 14 river basins is shown in Figure 3. It demonstrates that the dynamic BM model generally produces a 178 reasonably good fit to annual E for most of basins. The majority of arid basins with the aridity index > 1.0 (e.g. Inner 179 Mongolia, Qiangtang, Qinghai, Xinjiang, Hexi) have a high CE ranging between 0.70 and 0.97, and a small RE ranging 180 between -0.11% and 0.58%, indicating the performance of BM is rather reliable in arid basins at the annual scale. In contrast, 181 for humid basins with the aridity index

WSC is also used to model annual water balance. The improvement relative to the original BM in the simulated annual E can be observed from Figure 3 for the arid basins (with the aridity index > 1.0): the *CE* (*RE*) in these arid basins is ranging from 0.4–0.98 (-0.17% – 0.06%). However, virtually no improvement can be seen for the humid basins (Southeast Drainage, Pearl, and Yangtze basins) in which the performance (*CE*) becomes worse than the BM. Therefore, the extended BM may not provide significant improvement over the BM when applied to humid basins at the annual scale.

## 191 3.2.2 Grid scale

Similar to the basin scale, the BM for annual water balance modeling is applied to the 3814 grids over the entire China 193 (Figure 4). As seen from Figure 4a, there are 81%, 63.7% and 37.3% of grids with CE greater than 0, 0.5, and 0.8, 194 respectively. The RE for all grids ranges between -16.1% and 8.2% (Figure 4b). For the grids with the aridity index  $\leq$  3, CE 195 is considerably small under humid conditions and increases rapidly toward nearly 1.0 with the increasing aridity index 196 (Figure 4c). Similarly, RE is relatively large (small) under the humid (extremely dry) conditions (Figure 4e). Figure 4g 197 compares the observed and simulated interannual variability of E (as quantified by the standard deviation of annual E) for 198 the 3814 grids with the mean annual aridity index also given in the color bar. All the grids have the interannual variability of 199 E less than 120 mm. The correlation coefficient between observed and simulated interannual variability of E for all grids is 200 0.819, but the simulated E interannual variability is generally higher than observations (Figure 4g). Although performing 201 well for most grids in dry regions, the BM performs rather poorly for a large portion of grids in humid regions mainly due to 202 the impacts of interannual variability of WSC which have not been included in the BM.

The validation of the extended BM against the observed *E* (red line in Figure 4a) indicates that 83.5%, 70.1% and 48.1% of grids with the CE > 0, 0.5 and 0.8, respectively. The *RE* ranges between -16.1% to 6.9% (Figure 4b) rather close to the performance of BM. The correlation coefficient between observed and simulated interannual variability of *E* is 0.835, slightly larger than 0.819 of the BM. However, the overestimated interannual variability of *E* in BM in humid grids is significantly improved by the extended BM (Figure 4h). Overall the extended BM provides potential improvements of annual water balance at the grid scale mainly in correcting the overestimated interannual variability of *E*, but the improvement is limited for the grids with the underestimated interannual variability (Figures 4g and 4h).

#### 211 **3.3 Seasonal water balance**

## 212 **3.3.1 Basin scale**

Following the definition of the wet- and dry-season (equations (2) and (3)), the wet and dry seasonal  $PET/P_{e}, E/P_{e}$ 214 and  $E/P_e$  are calculated respectively in each basin. For some extremely humid (arid) basins, the mean monthly aridity indices 215 can be all smaller (larger) than 1.0 for all twelve months so that there is only wet (dry) season without any dry (wet) season. 216 Figure 5 shows the validation of the BM and extended BM against observed E at the dry seasonal scale for all 13 basins 217 except the Southeast Drainage basin which does not have any dry season. As seen, the BM performs poorly in capturing the 218 peaks and troughs of E in the Pearl, Yangtze, Southwest Drainage and Huaihe basins, with the CE all below 0.4. For the 219 other 9 less humid basins in Figure 5, the simulated E by the BM agrees well with observed E with the CE (RE) ranging 220 from 0.52 to 0.97 (0.02% and 0.24%). Significant improvements in the extended BM simulations are found for all basins, 221 particularly notable in the Pearl, Southwest Drainage, Huaihe and Liaohe basins. The CE of the extended BM ranges 222 between 0.44–0.98, and there are seven out of 13 basins with CE > 0.9, indicating that the extended BM simulation is 223 generally accurate in the dry seasonal scale.

The validations of the BM and extended BM against observed *E* in the wet seasonal scale are shown in Figure 6 for the only 8 out of the 14 basins with the wet season. As seen, the BM is unable to simulate *E* accurately with all *CE* less than 0.5. Similar to that found in the BM simulation, the extended BM simulations still show large discrepancy in reproducing the observed interannual variability of *E* at the wet seasonal scale. Particularly for the Huaihe basin, both the BM and the extended BM significantly underestimate the long-term mean *E*, with the *RE* –10.93% and –10.33% respectively. The simulated *E* cannot well capture the sharp rises and falls of interannual variability of *E*, likely due to the strong variability of water balance at the wet seasonal scale.

#### 232 **3.3.2 Grid scale**

Similar to the basin scale, the *PET/P*, *PET/P*<sub>e</sub>, *E/P* and *E/P*<sub>e</sub> are calculated separately for wet- and dry-season at the grid scale. The validation of the BM and extended BM against observed *E* in dry season for the selected grids is shown in Figure 7. There are 3538 out of the total 3814 grids (92.8%) with dry season identified within twelve months. For the performance

of BM, the CE is larger than 0, 0.5 and 0.8 for 83.1%, 64.5% and 39.9% of grids, respectively. The CE is small in humid 237 grids and increases as the aridity index increases. When moving toward increasingly drier climates, the CE gradually 238 becomes insensitive to the aridity index (> 20) and approaches to the upper limit of 1.0. Strong correlation (~0.94) is found 239 between the observed and simulated interannual variability of E in dry season (Figure 7g). However, the BM tends to 240 overestimate the interannual variability of E particularly for the humid basins with the RE as large as 124.3% (Figure 7e). 241 The results suggest that the BM is more applicable to arid than humid climates, similar to that found in analyzing annual 242 water balance. The validation of the extended BM performance against observed E indicates a significant improvement over 243 the BM. There are more than 96.3%, 85.4%, and 60% of grids with CE > 0, 0.5 and 0.8, respectively. The RE in humid 244 climates is reduced from >100% in BM to only about 10-20% in the extended BM (Figure 7f), also the overestimated 245 variability of simulated E in humid climates is largely corrected by the extended BM (Figure 7h) to achieve excellent 246 performance in water balance modeling in dry season.

The validation of the BM and extended BM against observed E in the wet season at the grid scale is shown in Figure 8. 249 There are in total 1982 out of 3814 grids (52%) with the wet season identified within twelve months. For these 1982 grids, 250 the CE of the BM simulation are all < 0.73, and only 41.3% of grids with CE > 0. The RE of the BM simulation is negative 251 for 83.4% of grids, and it tends to perform even worse in more arid climates (Figure 8e). Based on these results, it can be 252 concluded that the BM performance is not satisfactory in the wet season at the grid scale, similar to that found at the basin 253 scale. The performance of the extended BM in the wet season is similarly poor as the BM for most grids. Therefore, more 254 sophisticated water balance models with the capability of simulating important mechanisms in wet season are required to 255 model the interannual variability of water balance more accurately for all basins and grids under diverse climates.

**3.4 Monthly water balance** 

#### **3.4.1 Basin scale**

The BM and the extended BM are applied to model monthly water balance for the 14 river basins. The comparisons between the observed and simulated monthly *E* from both models are presented in Figure 9. Overall, the BM performs well in all basins with the *RE* (*CE*) between -16.6% and 9.5% (0.69 and 0.98). However, the BM tends to underestimate the peaks of *E*