# Peer review of "Modeling water balance using the Budyko framework over variable"

_Hydrology and Earth System Sciences, 2017_

## Referee Comment (RC1) · Anonymous Referee #1 · 15 Sep 2017

The aim of the study is to investigate the suitability of the Budyko framework under different timescales. It seems tempting to apply the simple Budyko model outside the steady state conditions, which are required to derive the water and energy limits; and many authors tried this (Zhang et al., 2008 J.Hydrol., Chen et al., 2013 WRR, Zeng and Cai 2014,2015, Greve et al., 2016 HESS, ...). The application at seasonal time scale requires modelling of water storage - which cannot be measured. Therefore the authors use model output from a land surface model to evaluate the Budyko model. After reading this - it is only mentioned in section 2 - I was tempted to reject the paper right away. This means the whole study is a simple model to model comparison study without even an assessment of the model output with real data. The authors evaluate

the model response of 14 large river basins but again only with the model output. The authors find what other have found as well, at shorter timescale model complexity must be larger. Yet, one finding which I find indeed interesting, is that there seems to be a relationship with the variability in water storage change and the error of using a simple Budyko type of model, shown in Figure 11. That means the larger the water storage changes in a catchment the larger the error using Budyko. Additional analysis on why this is case would be of interest, but is not provided. I guess it follows the reasoning of Zeng and Cai 2015 GRL highlighting the role of terrestrial water storage changes on ET.

I believe the manuscript must be improved in several ways to be of scientific significance.

First is to state right from the beginning that model output is being used to evaluate the Budyko model.

Second is to use actual runoff data of the large river basins to evaluate the Budyko model. Aggregated land surface model output is not very useful to my mind.

Third is an proper statistical assessment of the statement, that the Budyko models works better in arid than in humid areas.

4. the number of figures must be reduced. Some figures may be put into a supplement. One should select one timescale of the type of figures shown in Figures 4-10. Then a table of relevant statistics can be used instead.

5. Figures 11-13 show the same data with different ratios. I recommend to use Fig 11 and cut the remaining ones.

6. Use a better color scaling of the points such that the points get sufficiently different in color.

7. Use similar x and y scales in the panels of a Figure to allow fair visual comparison.

8. The discussion is rather a results section. Move the sections 4 to results. In the discussion please argue why the variability in water storage affects the model performance. Why the Budyko model is better in arid catchments. Please include and discuss your findings with the ones from the literature.

---

## Referee Comment (RC2) · Anonymous Referee #2 · 28 Sep 2017

The authors assess the application of the Budyko framework at short-time scales (monthly, seasonal and annual) in several major basins in China. They study steady-state conditions at these time-scales and the implications of water storage change on estimates of evapotranspiration by comparing estimates from the Fu equation and the "expanded" Fu (with effective precipitation) to "observations". They find that the Fu and expanded Fu models show important estimation improvements in arid basins but not in humid basins. They also note that estimates deteriorate at shorter time scales (monthly and wet-seasonal scale). First, I must say that the article is well written, consequent and explicative. The main objectives of the study also agree with current trends in Budyko framework research. However, I am concerned by one technical issue and

also the novelty of the work which should be better justified. I also think that the article could benefit from changes in structure that no make the Introduction too broad and the results too repetitive.

1. The authors compare the estimates of actual evapotranspiration from the Fu and expanded Fu (with effective precipitation) models (i.e., which they called the modelled results) with a product that they call "observations". My question is if this product really comprises "observations". From what I see, the "observations" are really "a model simulation forced by daily climate". Therefore, in reality, you are not comparing the Budyko-type models to observations but rather to a processed-based model. This needs to be clarified from the Abstract to the Conclusions. This is why you are getting extremely good model performances. If you really compared the two models with runoff observations, your results would show much worse model performances. So please justify the use of the VIC (Zhang, et al, 2014) data set as the main source to be compared with. Zhang et al. even say their ET estimates are just "reasonable". So maybe a good improvement to this article would be at least for the 14 basin scales to add the same analysis but with "real" runoff observations. 2. You have to be more specific in the way that your study is novel and unique (L. 72-78). There are many studies that are also studying water storage change and its implications within the Budyko framework. What novelty is you study presenting that it is not included in the references you mention (L. 87-99)? Now, it is not clear. Maybe you could be more specific, for example, regarding the specific case of China. 3. Literature review: I see important recent articles of water storage analysis and its implications for steady-state conditions in the Budyko framework missing from the Introduction. These also use both long and short-term assumptions on water storage change. Maybe they could benefit the introduction and discussion of this manuscript:

 c Gudmundsson, L., Greve, P. and Seneviratne, S. I.: The sensitivity of water availability to changes in the aridity index and other factors—A probabilistic analysis in the Budyko space, Geophys. Res. Lett., 2016GL069763, doi:10.1002/2016GL069763,

2016. • Jaramillo, F. and Destouni, G.: Developing water change spectra and distinguishing change drivers worldwide, Geophys. Res. Lett., 41(23), 8377–8386, doi:10.1002/2014GL061848, 2014. • Moussa, R. and Lhomme, J.-P.: The Budyko functions under non-steady-state conditions, Hydrol Earth Syst Sci, 20(12), 4867–4879, doi:10.5194/hess-20-4867-2016, 2016. • Ye, S., Li, H.-Y., Li, S., Leung, L. R., Demissie, Y., Ran, Q. and Blöschl, G.: Vegetation regulation on streamflow intra-annual variability through adaption to climate variations, Geophys. Res. Lett., 42(23), 2015GL066396, doi:10.1002/2015GL066396, 2015

4. The structure becomes too repetitive in the results and extended when jumping from multi-annual, annual, seasonal, monthly. Figure 3, 5 and 9 have to be simplified. Maybe just choose two example basins per each and discuss them thoroughly, and put the rest in Supplementary Information.

Other issues:

L. 30 "change in" soil water storage

L. 65 You are missing the reference study here.

L. 78 What do you mean by the "modelling errors"

L. 87 More details are needed concerning the VIC model since it is a core product used in your study.

L. 112 Citation?

L. 156 What is RE? You need to explain what this ratio represents and how it relates to model performance.

L. 162 replace "adequately" by "captures most of the"

L. 163 Again, I do not think calling this "observed" is adequate

L. 165-166 Where did you get this observation from?

Fig 4 and 6 Why giving so much emphasis to PET/P>10, they are extremely unique. Most of the grid points are PET/P<10 or even PET/P<5 so I would adjust the scale. Remember that PET/P>1 is still considered dry or arid so there is more to talk about in this range.

L. 187 Which basins are these.

L. 204-205 I could not get what you are referring to here.

What you all "Discussion" is still Results. You are missing the Discussion were you compare your results with other previous work on the field. See the references you are citing.

---

## Author Comment (AC1) · 17 Nov 2017

Dear Referee #1,

Thank you very much for the constructive comments concerning our manuscript entitled "Modeling water balance using the Budyko framework over variable timescales under diverse climates" (Manuscript No.: hess-2017-441). Your comments to our manuscript are all valuable. We have studied carefully and incorporated all of them into our revised manuscript. The point-to-point responses to your comments are given below.

**\*\*\*\*\*\*\*\*\*\*\*\*\*\*\*\*\*\*\*\*\*\*\*\*\*\*\*\*\*\*\*\*\*\*\*\*\*\*\*\*\*\*\*\*\*\*\*\*\*\*\*\*\*\*\*\*\*\*\*\*\*\*\*\*\*\*\*\*\*\*\*\***

**Comments from Anonymous Referee #1:**

The aim of the study is to investigate the suitability of the Budyko framework under different timescales. It seems tempting to apply the simple Budyko model outside the steady state conditions, which are required to derive the water and energy limits; and many authors tried this (Zhang et al., 2008 J.Hydrol., Chen et al., 2013 WRR, Zeng and Cai 2014,2015, Greve et al., 2016 HESS, ...). The application at seasonal time scale requires modelling of water storage - which cannot be measured. Therefore the authors use model output from a land surface model to evaluate the Budyko model. After reading this - it is only mentioned in section 2 - I was tempted to reject the paper right away. This means the whole study is a simple model to model comparison study without even an assessment of the model output with real data. The authors evaluate the model response of 14 large river basins but again only with the model output. The authors find what other have found as well, at shorter timescale model complexity must be larger. Yet, one finding which I find indeed interesting, is that there seems to be a relationship with the variability in water storage change and the error of using a simple Budyko type of model, shown in Figure 11. That means the larger the water storage changes in a catchment the larger the error using Budyko. Additional analysis on why this is case would be of interest, but is not provided. I guess it follows the reasoning of Zeng and Cai 2015 GRL highlighting the role of terrestrial water storage changes on ET.

**Response:** Thank you for all the constructive comments. In this study, we investigated the suitability of the Budyko model (BM) at various timescales in 14 major basins in China based the VIC model output data.

We recognize that there are several previous studies related to examine the effect of water storage change (*WSC*) using the Budyko framework. In fact, our study differs from those previous studies in the following aspects: (1) we test the application of the BM at diverse climates (arid and humid climates) with a very wide range of the aridity index, and address the important question regarding the accuracy of BM when applied to different climates (2) we provide quantitative assessment of the influence of *WSC* on the performance of BM under both arid and humid climates; (3) we use the entire China comprising 14 large river basins as our study regions.

I believe the manuscript must be improved in several ways to be of scientific significance.

**1.** First is to state right from the beginning that model output is being used to evaluate the Budyko model.

**Response:** Thank for your comment. Following your suggestions, we have stated clear at the beginning of our revised manuscript (Abstract and Introduction) on using model output data for evaluating the Budyko framework.

**2.** Second is to use actual runoff data of the large river basins to evaluate the Budyko model. Aggregated land surface model output is not very useful to my mind.

**Response:** Thank for your comment. In this study, we assessed the application of the Budyko model (BM) at five various timescales (mean-annual, annual, dry-season, wet-season and monthly) in 14 major river basins in China based on the output data of the VIC model. According to Zhang et al (2014), the VIC model has been calibrated and validated in the 11 major river basins of China based on the observed hydrologic data of 15 hydrologic stations. The model parameters were estimated by using the optimization algorithm of the multi-objective complex evolution of the University of Arizona (MOCOM-UA). The results indicated that the simulated monthly runoff matches well with observations - the Nash–Sutcliffe efficiency is larger than 0.8 at 11 out of the total 15 stations, and the relative error is less than 25% at 13 out of 15 stations. Only for the stations at the Tibetan Plateau of western China, there are relatively large errors in hydrologic simulation due to lack of data with sufficiently reliable quality. In addition, the VIC-simulated evapotranspiration ($E$) was compared well with a global product of $E$ based on the multi-sensor remote sensing ($RS$) data with the relative difference less than 25% over the most areas of China, and the relative bias of $E$ averaged over the entire China is only – 4%, indicating satisfactory simulation of $E$ by VIC over China.

In our study, the BM was applied at five different timescales, namely, the mean-annual, annual, dry- and wet-seasonal, and monthly timescales. At the mean-annual scale, the data of precipitation ($P$), runoff ($R$), and potential evapotranspiration ($PET$) are used to fit the Fu's equation (($P$-$R$)/$P$ vs. $PET$/$P$) assuming that water storage change ($WSC$) is negligible at the mean-annual timescale. However, at the annual and even finer timescales, the VIC-simulated evapotranspiration ($E$) data are used to fit the Fu's equations instead of using ($P$-$R$), thus the effects of $WSC$ is considered. In assessing the original BM, the VIC-simulated $R$ was only used at the mean-annual timescale, but in the extended BM the effective $P$ is calculated as $P - WSC = P - (P - E - R) = E + R$. Therefore, VIC-simulated $E$ and $R$ are both used to fit the extended BM at the annual and finer timescales.

We recognize that the use of modeling data may introduce uncertainty into the assessment of BM. However, we were unable to find observed runoff (streamflow) data in the 14 river basins considered due to the general difficulty in acquiring Chinese streamflow data, which are confidential materials only kept by government authorities. Also, there is generally a lack of a long-term observed runoff data in the west large basins of China (e.g., Xinjiang and Hexi).

However, following your review comments, we made efforts to obtain observed streamflow data at 14 hydrologic stations (as plotted and summarized in Figure S1 and Table S1) located at 7 large river basins of China (i.e., the Southeast Drainage, Southwest Drainage, Yangtze, Yellow, Huaihe, Heilongjiang, and Liaohe basin), with the drainage areas ranging from $19.1 \times 10^3$ to $1705.4 \times 10^3$ km$^2$. Based on these observed streamflow data at 14 sub-basins, we conducted an assessment of the

performance of BM and extended BM at the annual and monthly timescales (Table S2). As shown in Table S2, the performance of BM decreases when moving from arid to humid basins at all four timescales. In general, the BM predicts a reasonably well $E$ for most arid basins (e.g., from Haerbin to Tieling) at the annual, dry-seasonal, and monthly timescales, with the $CE$ ranging between 0.56~0.95 and the $RE$ between -4.60%~2.43%. In contrast, a poor performance is found in the relatively humid basins, particularly at the wet-seasonal timescale where $CE$ is generally negative. When $WSC$ is incorporated into BM (by replacing $P$ with effective $P$), the improvement is found in arid basins, but to a lesser extent in humid basins (e.g., Chaoan, Gaoyao and Datong). For example, at the monthly timescale both BM and extended BM underestimate $E$ at almost all the basins, and this underestimation is more serious for the extended BM. In addition, for the Wujiadu station in the Huaihe basin, both BM and extended BM cannot well reproduce the variability of $E$ ($CE$ is small or negative), which is similar to the finding in the Huaihe basin by the VIC-simulated data. Overall, the results of the assessment of BM and extend BM with the observed runoff data at the 14 sub-basins are consistent with the results obtained from the assessment with the VIC-simulated runoff data.

Following your comments, we add two sub-sections in our revised manuscript for assessing the water balance modeling of the Budyko framework with (1) the VIC-simulated runoff data (section 3.1) and with (2) the observed runoff data at the 14 sub-basin hydrologic stations (section 3.2).

Reference
Zhang, X., Tang, Q., Pan, M., Tang, Y., 2014. A Long-Term Land Surface Hydrologic Fluxes and States Dataset for China. J. Hydrometeor. 15, 2067–2084, doi: 10.1175/JHM-D-13-0170.1.

[Figure]

Figure S1. Locations of the 14 large river basins in China and the associated 14 sub-basin hydrological stations: 1, Southeast Drainage; 2, Pearl River; 3, Yangtze River; 4, Southwest Drainage; 5, Huaihe River; 6, Heilongjiang River; 7, Liaohe River; 8, Haihe River; 9, Yellow River; 10, Inner Mongolia River; 11, Qiangtang River; 12, Qinghai River; 13, Xinjiang River, 14, Hexi River.

Table S1. List of the 14 sub-basin hydrologic stations.

| Name | Basin | Drainage area (km$^2$) | Lat ($^o$ N) | Lon ($^o$ E) | Period | Aridity index |
|------|-------|------------------------|--------------|--------------|--------|---------------|
| Chaoan | Southeast Drainage | 29077 | 23.67 | 116.65 | 1980-2008 | 0.55 |
| Gaoyao | Southeast Drainage | 351535 | 23.05 | 112.47 | 1980-2008 | 0.69 |
| Datong | Yangtze | 1705383 | 30.78 | 117.61 | 1980-2008 | 0.82 |
| Wujiadu | Huaihe | 121330 | 32.96 | 117.38 | 1980-2008 | 1.05 |
| Yunjinghong | Southwest Drainage | 141779 | 22.03 | 100.79 | 1980-2007 | 1.15 |
| Jiamusi | Heilongjiang | 528277 | 46.82 | 130.37 | 1981-2002 | 1.36 |
| Pingshan | Yangtze | 485099 | 28.64 | 104.17 | 1980-2008 | 1.37 |
| Haerbin | Heilongjiang | 390526 | 45.77 | 126.59 | 1981-2002 | 1.48 |
| Weijiabao | Yellow | 37037 | 34.29 | 107.73 | 1980-2008 | 1.83 |
| Shangquan | Yellow | 182821 | 36.07 | 103.30 | 1987-2007 | 1.93 |
| Zhimenda | Yangtze | 137704 | 32.99 | 97.25 | 1980-2008 | 2.24 |
| Huayuankou | Yellow | 730036 | 34.91 | 113.67 | 1980-2007 | 2.54 |
| Xinglongpo | Liaohe | 19100 | 42.32 | 119.43 | 1980-2008 | 2.56 |
| Tieling | Liaohe | 120800 | 42.33 | 123.84 | 1992-2007 | 2.61 |

Table S2. Statistics on performance of the original BM (O-BM) and extended BM (E-BM) in water balance modeling for the 14 hydrological stations at four different timescales.

| No. | Stations | Annual | | | | Dry season | | | | Wet season | | | | Month | | | |
|-----|----------|--------|------|--------|------|------------|------|--------|------|------------|------|--------|------|-------|------|--------|------|
| | | *CE* | | *RE* (%) | | *CE* | | *RE* (%) | | *CE* | | *RE* (%) | | *CE* | | *RE* (%) | |
| | | O-BM | E-BM | O-BM | E-BM | O-BM | E-BM | O-BM | E-BM | O-BM | E-BM | O-BM | E-BM | O-BM | E-BM | O-BM | E-BM |
| 1 | Chaoan | -0.22 | -0.58 | 0.06 | 0.50 | 0.08 | 0.46 | 7.77 | -3.04 | -0.75 | -0.65 | -1.87 | -1.50 | 0.79 | 0.50 | -8.28 | -25.39 |
| 2 | Gaoyao | 0.02 | 0.08 | -0.05 | -0.12 | 0.65 | 0.53 | 0.35 | -0.18 | -2.79 | -2.79 | -4.68 | -4.69 | 0.74 | 0.61 | -14.20 | -23.33 |
| 3 | Datong | 0.66 | 0.51 | -0.09 | -0.01 | 0.42 | 0.36 | 0.31 | 0.14 | 0.41 | 0.38 | -0.33 | -0.26 | 0.75 | 0.67 | -15.98 | -20.93 |
| 4 | Wujiadu | -0.70 | -0.52 | 1.08 | 0.92 | -2.33 | -0.88 | 2.39 | 1.02 | -8.67 | -8.28 | -13.24 | -12.45 | 0.65 | 0.68 | -13.66 | -24.61 |
| 5 | Yunjinghong | 0.38 | 0.64 | -0.02 | 0.03 | 0.23 | 0.64 | 0.36 | -0.02 | 0.09 | 0.49 | -0.07 | -0.20 | 0.94 | 0.71 | -0.39 | -20.63 |
| 6 | Jiamusi | 0.83 | 0.06 | -0.19 | 0.05 | 0.83 | 0.42 | 0.08 | -0.07 | -0.18 | -1.05 | -4.33 | -2.76 | 0.96 | 0.88 | 0.37 | -11.88 |
| 7 | Pingshan | 0.40 | 0.57 | 0.05 | 0.11 | 0.08 | 0.61 | 0.30 | 0.05 | 0.23 | 0.61 | 0.02 | -0.06 | 0.98 | 0.69 | 1.19 | -24.74 |
| 8 | Haerbin | 0.89 | 0.16 | -0.15 | 0.05 | 0.85 | 0.51 | 0.20 | -0.10 | 0.01 | -0.86 | -4.12 | -2.41 | 0.95 | 0.87 | -0.13 | -13.42 |
| 9 | Weijiabao | 0.78 | 0.76 | 0.28 | 0.21 | 0.78 | 0.76 | 0.28 | 0.21 | - | - | - | - | 0.93 | 0.97 | -4.51 | -4.06 |
| 10 | Shangquan | 0.75 | 0.63 | 0.06 | -0.03 | 0.75 | 0.63 | 0.06 | -0.03 | - | - | - | - | 0.96 | 0.65 | -4.60 | -27.84 |
| 11 | Zhimenda | 0.56 | 0.86 | 0.23 | 0.06 | 0.56 | 0.86 | 0.23 | 0.06 | - | - | - | - | 0.95 | 0.84 | 2.43 | -20.79 |
| 12 | Huayuankou | 0.83 | 0.84 | 0.05 | 0.04 | 0.83 | 0.84 | 0.05 | 0.04 | - | - | - | - | 0.97 | 0.89 | -3.78 | -14.22 |
| 13 | Xinglongpo | 0.72 | 0.92 | 0.33 | 0.00 | 0.72 | 0.92 | 0.33 | 0.00 | - | - | - | - | 0.93 | 0.99 | -2.02 | -0.21 |
| 14 | Tieling | 0.82 | 0.95 | 0.22 | 0.00 | 0.82 | 0.95 | 0.22 | 0.00 | - | - | - | - | 0.95 | 0.99 | -4.29 | 2.44 |

**3.** Third is an proper statistical assessment of the statement, that the Budyko models works better in arid than in humid areas.

**Response:** Thank for your comment. According to McVicar et al. (2012), the grids over China are classified into three cases: the energy-limited (humid) areas (aridity index < 0.76), the water-limited (dry) areas (aridity index > 1.35), and the intermedium areas (0.76 < aridity index <1.35). According to your suggestions, we conducted a comparison of the cumulative distribution function (CDF) of the Coefficient of Efficiency (*CE*) and the Relative Error (*RE*) in the humid and dry grids over China at the annual, dry season, wet season, and monthly timescales. As shown in Figures S2 and S3, there are large systematic differences in the *CE* and *RE* of the BM between the humid and dry areas. Given a fixed *CE* within the range from 0 to 1, the CDF is significantly larger in humid areas than in dry areas at all four timescales, especially at the annual and dry season scales. Given a fixed *RE*, the CDF is smaller in humid areas than in dry areas at the annual, dry-season and monthly scales (except for the wet-season scale). These results indicate that the BM works better in the arid than in humid areas at the annual, dry-season, and monthly timescales. According to your suggestion, in the revised manuscript we added a new subsection (section 3.3) within the Results section to make a comparison of the model performance in humid and arid regions.

[Figure]

Figure S2. The cumulative distribution function (CDF) of *CE* in the humid grids (with the aridity index < 0.76) and the dry grids (with the aridity index > 1.35) at the (a) annual, (b) dry-season, (c) wet-season, and (d) monthly timescales. Dotted lines represent 95% confidence intervals.

[Figure]

Figure S3. The cumulative distribution function (CDF) of the absolute values of *RE* (%) in the humid grids (with the aridity index < 0.76) and the dry grids (with the aridity index > 1.35) at the (a) annual, (b) dry-season, (c) wet-season, and (d) monthly scales. Dotted lines represent 95% confidence intervals.

Reference:

McVicar, T.R., Roderick, M.L., Donohue, R.J., Van Niel, T.G., 2012. Less bluster ahead? Ecohydrological implications of global trends of terrestrial near-surface wind speeds. Ecohydrol. 5, 381–388, doi: 10.1002/eco.1298.

**4.** the number of figures must be reduced. Some figures may be put into a supplement. One should select one timescale of the type of figures shown in Figures 4-10. Then a table of relevant statistics can be used instead.

**Response:** Thank for your comment. According to your suggestions, we have reduced 8 figures (e.g., the original Figures 5-10 and Figures 12 and 13). Two tables summarizing relevant statistics are used to replace the original Figures 5-10 in the revised manuscript.

**5.** Figures 11-13 show the same data with different ratios. I recommend to use Fig 11 and cut the remaining ones.

**Response:** Thank for your comment. We have removed the original Figures 12 and 13 and only keep the original Figure 11 in the revised manuscript.

**6.** Use a better color scaling of the points such that the points get sufficiently different in color.

**Response:** Thank for your comment. We have revised it.

**7.** Use similar x and y scales in the panels of a Figure to allow fair visual comparison.

**Response:** Thank for your comment. We have revised it.

**8.** The discussion is rather a results section. Move the sections 4 to results. In the discussion please argue why the variability in water storage affects the model performance. Why the Budyko model is better in arid catchments. Please include and discuss your findings with the ones from the literature.

**Response:** Thank for your comment. According to your suggestion, we have moved the Section 4 to the Results section in the revised manuscript. In the Discussion section, we focused on the following two issues: (1) why the Budyko model performs better in the arid catchments? and (2) how the variability in the water storage affects model performance? Meanwhile, we also provide discussion on the comparison of our findings with previous studies in literature.

---

## Author Comment (AC2) · 17 Nov 2017

Dear Referee #2,

Thanks very much for your constructive comments concerning our manuscript entitled "Modeling water balance using the Budyko framework over variable timescales under diverse climates" (Manuscript No.: hess-2017-441). All your comments are valuable and helpful for us to revise our paper. We have studied all your comments carefully and replied you point-by-point below.

**\*\*\*\*\*\*\*\*\*\*\*\*\*\*\*\*\*\*\*\*\*\*\*\*\*\*\*\*\*\*\*\*\*\*\*\*\*\*\*\*\*\*\*\*\*\*\*\*\*\*\*\*\*\*\*\*\*\*\*\*\*\*\*\*\*\*\*\*\*\*\*\*\*\*\*\*\*\***

**Comments from Anonymous Referee #2:**

The authors assess the application of the Budyko framework at short-time scales (monthly, seasonal and annual) in several major basins in China. They study steadystate conditions at these time-scales and the implications of water storage change on estimates of evapotranspiration by comparing estimates from the Fu equation and the "expanded" Fu (with effective precipitation) to "observations". They find that the Fu and expanded Fu models show important estimation improvements in arid basins but not in humid basins. They also note that estimates deteriorate at shorter time scales (monthly and wet-seasonal scale). First, I must say that the article is well written, consequent and explicative. The main objectives of the study also agree with current trends in Budyko framework research. However, I am concerned by one technical issue and also the novelty of the work which should be better justified. I also think that the article could benefit from changes in structure that no make the Introduction too broad and the results too repetitive.

Response: Thank for your comment. We have incorporated all your comments in the revised manuscript. Below please find our point-to-point responses to each of your comments.

**1.** The authors compare the estimates of actual evapotranspiration from the Fu and expanded Fu (with effective precipitation) models (i.e., which they called the modelled results) with a product that they call "observations". My question is if this product really comprises "observations". From what I see, the "observations" are really "a model simulation forced by daily climate". Therefore, in reality, you are not comparing the Budyko-type models to observations but rather to a processed-based model. This needs to be clarified from the Abstract to the Conclusions. This is why you are getting extremely good model performances. If you really compared the two models with runoff observations, your results would show much worse model performances. So please justify the use of the VIC (Zhang, et al, 2014) data set as the main source to be compared with. Zhang et al. even say their ET estimates are just "reasonable". So maybe a good improvement to this article would be at least for the 14 basin scales to add the same analysis but with "real" runoff observations.

Response: Thank for your comment. In this study, we assessed the application of the Budyko model (BM) at five various timescales (mean-annual, annual, dry-seasonal, wet-seasonal and monthly) at 14 major river basins of China based on the modeling output data of the VIC land surface model. We fully agree that our paper uses model-simulated data to compare with the prediction from the BM, so we did not use "observations" in this study. This important point has been clearly clarified throughout the revised manuscript.

According to Zhang et al (2014), the VIC model has been calibrated and validated in the 11 major river basins of China based on the observed hydrologic data of 15 hydrologic stations. The model parameters were estimated by using the optimization algorithm of the multi-objective complex evolution of the University of Arizona (MOCOM-UA). The results indicated that the simulated monthly runoff matches well with observations - the Nash–Sutcliffe efficiency is larger than 0.8 at 11 out of the total 15 stations, and the relative error is less than 25% at 13 out of 15 stations. Only for the stations at the Tibetan Plateau of western China, there are relatively large errors in hydrologic simulation due to lack of data with sufficiently reliable quality. In addition, the VIC-simulated evapotranspiration ($E$) was compared well with a global product of $E$ based on the multi-sensor remote sensing ($RS$) data with the relative difference less than 25% over the most areas of China, and the relative bias of $E$ averaged over the entire China is only – 4%, indicating satisfactory simulation of $E$ by VIC over China.

In our study, the BM was applied at five different timescales, namely, the mean-annual, annual, dry- and wet-seasonal, and monthly timescales. At the mean-annual scale, the data of precipitation ($P$), runoff ($R$), and potential evapotranspiration ($PET$) are used to fit the Fu's equation (($P$-$R$)/$P$ vs. $PET$/$P$) assuming that water storage change ($WSC$) is negligible at the mean-annual timescale. However, at the annual and even finer timescales, the VIC-simulated evapotranspiration ($E$) data are used to fit the Fu's equations instead of using ($P$-$R$), thus the effects of $WSC$ is considered. In assessing the original BM, the VIC-simulated $R$ was only used at the mean-annual timescale, but in the extended BM the effective $P$ is calculated as $P - WSC = P - (P - E - R) = E + R$. Therefore, VIC-simulated $E$ and $R$ are both used to fit the extended BM at the annual and finer timescales.

We recognize that the use of modeling data may introduce uncertainty into the assessment of BM. However, we were unable to find observed runoff (streamflow) data in the 14 river basins considered due to the general difficulty in acquiring Chinese streamflow data, which are confidential materials only kept by government authorities. Also, there is generally a lack of a long-term observed runoff data in the west large basins of China (e.g., Xinjiang and Hexi).

However, following your review comments, we made efforts to obtain observed streamflow data at 14 hydrologic stations (as plotted and summarized in Figure S1 and Table S1) located at 7 large river basins of China (i.e., the Southeast Drainage, Southwest Drainage, Yangtze, Yellow, Huaihe, Heilongjiang, and Liaohe basin), with the drainage areas ranging from $19.1 \times 10^3$ to $1705.4 \times 10^3$ km$^2$. Based on these observed streamflow data at 14 sub-basins, we conducted an assessment of the performance of BM and extended BM at the annual and monthly timescales (Table S2). As shown in Table S2, the performance of BM decreases when moving from arid to humid basins at all four timescales. In general, the BM predicts a reasonably well $E$ for most arid basins (e.g., from Haerbin to Tieling) at the annual, dry-seasonal, and monthly timescales, with the $CE$ ranging between 0.56~0.95 and the $RE$ between -4.60%~2.43%. In contrast, a poor performance is found in the relatively humid basins, particularly at the wet-seasonal timescale where $CE$ is generally negative. When $WSC$ is incorporated into BM (by replacing $P$ with effective $P$), the improvement is found in arid basins, but to a lesser extent in humid basins (e.g., Chaoan, Gaoyao and Datong). For example, at the monthly timescale both BM and extended BM underestimate $E$ at almost all

the basins, and this underestimation is more serious for the extended BM. In addition, for the Wujiadu station in the Huaihe basin, both BM and extended BM cannot well reproduce the variability of $E$ ($CE$ is small or negative), which is similar to the finding in the Huaihe basin by the VIC-simulated data. Overall, the results of the assessment of BM and extend BM with the observed runoff data at the 14 sub-basins are consistent with the results obtained from the assessment with the VIC-simulated runoff data.

Following your comments, we add two sub-sections in our revised manuscript for assessing the water balance modeling of the Budyko framework with (1) the VIC-simulated runoff data (section 3.1) and with (2) the observed runoff data at the 14 sub-basin hydrologic stations (section 3.2).

Reference

Zhang, X., Tang, Q., Pan, M., Tang, Y., 2014. A Long-Term Land Surface Hydrologic Fluxes and States Dataset for China. J. Hydrometeor. 15, 2067–2084, doi: 10.1175/JHM-D-13-0170.1.

[Figure]

Figure S1. Locations of the 14 large river basins in China and the associated 14 sub-basin hydrological stations: 1, Southeast Drainage; 2, Pearl River; 3, Yangtze River; 4, Southwest Drainage; 5, Huaihe River; 6, Heilongjiang River; 7, Liaohe River; 8, Haihe River; 9, Yellow River; 10, Inner Mongolia River; 11, Qiangtang River; 12, Qinghai River; 13, Xinjiang River, 14, Hexi River.

Table S1. List of the 14 sub-basin hydrologic stations.

| Name | Basin | Drainage area (km²) | Lat (° N) | Lon (° E) | Period | Aridity index |
|---|---|---|---|---|---|---|
| Chaoan | Southeast Drainage | 29077 | 23.67 | 116.65 | 1980-2008 | 0.55 |
| Gaoyao | Southeast Drainage | 351535 | 23.05 | 112.47 | 1980-2008 | 0.69 |
| Datong | Yangtze | 1705383 | 30.78 | 117.61 | 1980-2008 | 0.82 |
| Wujiadu | Huaihe | 121330 | 32.96 | 117.38 | 1980-2008 | 1.05 |

| Yunjinghong | Southwest Drainage | 141779 | 22.03 | 100.79 | 1980-2007 | 1.15 |
|---|---|---|---|---|---|---|
| Jiamusi | Heilongjiang | 528277 | 46.82 | 130.37 | 1981-2002 | 1.36 |
| Pingshan | Yangtze | 485099 | 28.64 | 104.17 | 1980-2008 | 1.37 |
| Haerbin | Heilongjiang | 390526 | 45.77 | 126.59 | 1981-2002 | 1.48 |
| Weijiabao | Yellow | 37037 | 34.29 | 107.73 | 1980-2008 | 1.83 |
| Shangquan | Yellow | 182821 | 36.07 | 103.30 | 1987-2007 | 1.93 |
| Zhimenda | Yangtze | 137704 | 32.99 | 97.25 | 1980-2008 | 2.24 |
| Huayuankou | Yellow | 730036 | 34.91 | 113.67 | 1980-2007 | 2.54 |
| Xinglongpo | Liaohe | 19100 | 42.32 | 119.43 | 1980-2008 | 2.56 |
| Tieling | Liaohe | 120800 | 42.33 | 123.84 | 1992-2007 | 2.61 |

Table S2. Statistics on the performance of the original BM (O-BM) and the extended BM (E-BM) for the 14 sub-basin hydrological stations for four different timescales.

| No. | Stations | Annual | | | | Dry season | | | | Wet season | | | | Month | | | |
|---|---|---|---|---|---|---|---|---|---|---|---|---|---|---|---|---|---|
| | | *CE* | | *RE* (%) | | *CE* | | *RE* (%) | | *CE* | | *RE* (%) | | *CE* | | *RE* (%) | |
| | | O-BM | E-BM | O-BM | E-BM | O-BM | E-BM | O-BM | E-BM | O-BM | E-BM | O-BM | E-BM | O-BM | E-BM | O-BM | E-BM |
| 1 | Chaoan | -0.22 | -0.58 | 0.06 | 0.50 | 0.08 | 0.46 | 7.77 | -3.04 | -0.75 | -0.65 | -1.87 | -1.50 | 0.79 | 0.50 | -8.28 | -25.39 |
| 2 | Gaoyao | 0.02 | 0.08 | -0.05 | -0.12 | 0.65 | 0.53 | 0.35 | -0.18 | -2.79 | -2.79 | -4.68 | -4.69 | 0.74 | 0.61 | -14.20 | -23.33 |
| 3 | Datong | 0.66 | 0.51 | -0.09 | -0.01 | 0.42 | 0.36 | 0.31 | 0.14 | 0.41 | 0.38 | -0.33 | -0.26 | 0.75 | 0.67 | -15.98 | -20.93 |
| 4 | Wujiadu | -0.70 | -0.52 | 1.08 | 0.92 | -2.33 | -0.88 | 2.39 | 1.02 | -8.67 | -8.28 | -13.24 | -12.45 | 0.65 | 0.68 | -13.66 | -24.61 |
| 5 | Yunjinghong | 0.38 | 0.64 | -0.02 | 0.03 | 0.23 | 0.64 | 0.36 | -0.02 | 0.09 | 0.49 | -0.07 | -0.20 | 0.94 | 0.71 | -0.39 | -20.63 |
| 6 | Jiamusi | 0.83 | 0.06 | -0.19 | 0.05 | 0.83 | 0.42 | 0.08 | -0.07 | -0.18 | -1.05 | -4.33 | -2.76 | 0.96 | 0.88 | 0.37 | -11.88 |
| 7 | Pingshan | 0.40 | 0.57 | 0.05 | 0.11 | 0.08 | 0.61 | 0.30 | 0.05 | 0.23 | 0.61 | 0.02 | -0.06 | 0.98 | 0.69 | 1.19 | -24.74 |
| 8 | Haerbin | 0.89 | 0.16 | -0.15 | 0.05 | 0.85 | 0.51 | 0.20 | -0.10 | 0.01 | -0.86 | -4.12 | -2.41 | 0.95 | 0.87 | -0.13 | -13.42 |
| 9 | Weijiabao | 0.78 | 0.76 | 0.28 | 0.21 | 0.78 | 0.76 | 0.28 | 0.21 | - | - | - | - | 0.93 | 0.97 | -4.51 | -4.06 |
| 10 | Shangquan | 0.75 | 0.63 | 0.06 | -0.03 | 0.75 | 0.63 | 0.06 | -0.03 | - | - | - | - | 0.96 | 0.65 | -4.60 | -27.84 |
| 11 | Zhimenda | 0.56 | 0.86 | 0.23 | 0.06 | 0.56 | 0.86 | 0.23 | 0.06 | - | - | - | - | 0.95 | 0.84 | 2.43 | -20.79 |
| 12 | Huayuankou | 0.83 | 0.84 | 0.05 | 0.04 | 0.83 | 0.84 | 0.05 | 0.04 | - | - | - | - | 0.97 | 0.89 | -3.78 | -14.22 |
| 13 | Xinglongpo | 0.72 | 0.92 | 0.33 | 0.00 | 0.72 | 0.92 | 0.33 | 0.00 | - | - | - | - | 0.93 | 0.99 | -2.02 | -0.21 |
| 14 | Tieling | 0.82 | 0.95 | 0.22 | 0.00 | 0.82 | 0.95 | 0.22 | 0.00 | - | - | - | - | 0.95 | 0.99 | -4.29 | 2.44 |

**2.** You have to be more specific in the way that your study is novel and unique (L. 72-78). There are many studies that are also studying water storage change and its implications within the Budyko framework. What novelty is you study presenting that it is not included in the references you mention (L. 87-99)? Now, it is not clear. Maybe you could be more specific, for example, regarding the specific case of China.

Response: Thank for your comment. We recognize that there are several previous studies related to examine the effect of water storage change (*WSC*) using the Budyko framework. In fact, our study differs from those previous studies in the following aspects: (1) we test the application of the BM at diverse climates (arid and humid climates) with a very wide range of the aridity index, and address the important question regarding the accuracy of BM when applied to different climates (2) we provide quantitative assessment of the influence of *WSC* on the performance of BM under both arid and humid climates; (3) we use the entire Chian comprising 14 large river basins as our study regions. Following your instruction, we have rewritten the sentences (L.72-78) to indicate these unique points of our study.

**3.** Literature review: I see important recent articles of water storage analysis and its implications for steady-state conditions in the Budyko framework missing from the Introduction. These also use both long and shortterm assumptions on water storage change. Maybe they could benefit the introduction and discussion of this manuscript:

âA ˘ c Gudmundsson, L., Greve, P. and Seneviratne, S. I.: The sensitivity of water availability to changes in the aridity index and other factorsâA ˘TA probabilistic analysis in the ˘ Budyko space, Geophys. Res. Lett., 2016GL069763, doi:10.1002/2016GL069763, 2016.
âA ˘ c Jaramillo, F. and Destouni, G.: Developing water change spectra and ´distinguishing change drivers worldwide, Geophys. Res. Lett., 41(23), 8377–8386, doi:10.1002/2014GL061848, 2014.
âA ˘ c Moussa, R. and Lhomme, J.-P.: The Budyko ´functions under non-steady-state conditions, Hydrol Earth Syst Sci, 20(12), 4867–4879, doi:10.5194/hess-20-4867-2016, 2016.
âA ˘ c Ye, S., Li, H.-Y., Li, S., Leung, L. ´R., Demissie, Y., Ran, Q. and Blöschl, G.: Vegetation regulation on streamflow intraannual variability through adaption to climate variations, Geophys. Res. Lett., 42(23), 2015GL066396, doi:10.1002/2015GL066396, 2015

Response: Thank for your comment. These references are very helpful for revising our paper. We have studied them and cited these references in our revised manuscript to provide more comprehensive literature view on the impact of *WSC* within the Budyko framework.

**4.** The structure becomes too repetitive in the results and extended when jumping from multi-annual, annual, seasonal, monthly. Figure 3, 5 and 9 have to be simplified. Maybe just choose two example basins per each and discuss them thoroughly, and put the rest in Supplementary Information.

Response: Thank for your comment. We completely agree that the structure of our manuscript is too repetitive in the presentation of our results. Following your suggestions from and also another anonymous referee #1, we have deleted the repetitive figures (e.g., the original Figures 5-10 and

Figures 12 and 13) and just keep the figure for the results at the annual timescale. The results for other four timescales are summarized in two Tables in the revised manuscript to replace Figures 5-10 in the original manuscript.

**Other issues:**

L. 30 "change in" soil water storage

Response: Thank for your comment. We have revised it.

L. 65 You are missing the reference study here.

Response: Thank for your comment. We have added the reference there.

L. 78 What do you mean by the "modelling errors"

Response: Thank for your comment. We have changed "modelling errors" into "prediction errors" in the revised manuscript.

L. 87 More details are needed concerning the VIC model since it is a core product used in your study.

Response: Thank for your comment. We have added more detailed information about the VIC model simulation and its calibration and validation in the revised manuscript.

L. 112 Citation?

Response: Thank for your comment. We added the URL (http://www.resdc.cn/) for accessing the NPP data of the Research Center for Natural resources and Environment Science Data of the Chinese Academy of Science.

L. 156 What is RE? You need to explain what this ratio represents and how it relates to model performance.

Response: Thank for your comment. "*RE*" stands for the relative error, which is the percentage difference of BM prediction relative to observations. The BM prediction is more accurate as the absolute value of *RE* decreases. We have added more information about *RE* in the revised manuscript according to your suggestions.

L. 162 replace "adequately" by "captures most of the"

Response: Thank for your comment. We have revised it.

L. 163 Again, I do not think calling this "observed" is adequate

Response: Thank for your comment. We agree that it is not adequate to call the "observed" here. Therefore, the VIC model output is renamed as "model-simulated" in the revised manuscript, so the sentence has been changed to "……shows the comparison between the model-simulated and the BM-predicted mean annual *E* in the 14 river basins.

L. 165-166 Where did you get this observation from?

Response: Thank for your comment. We have removed the "observation" from the sentence and

changed the sentence to: "…indicating that the *E* is well estimated by BM at the long-term mean annual timescale".

Fig 4 and 6 Why giving so much emphasis to PET/P>10, they are extremely unique. Most of the grid points are PET/P<10 or even PET/P<5 so I would adjust the scale. Remember that PET/P>1 is still considered dry or arid so there is more to talk about in this range.

Response: Thank for your comment. We have decreased the x scale in the panels of the Figures and just keep the grid points with PET/P<10, which is a more common range of this aridity index.

L. 187 Which basins are these.

Response: Thank for your comment. 'these' refers to the dry basins (with the aridity index > 1.0). We have rewritten the sentence in the revised manuscript.

L. 204-205 I could not get what you are referring to here

Response: Thank for your comment. In the revised manuscript, a new table is added to include the statistical information for the percentage of total grids where the Coefficient of Efficiency (*CE*) of BM predictions exceeds 0, 0.5, and 0.8 (L. 204-205).

What you all "Discussion" is still Results. You are missing the Discussion were you compare your results with other previous work on the field. See the references you are citing.

Response: Thank for your comment. According to the suggestion from you and Referee #1, we have rewritten the Discussion section to compare our findings with previous studies in literature. Particularly, we focused on the following two issues: (1) why the Budyko model performs better in the arid catchments? and (2) how the variability in the water storage affects model performance?